# Antibiotic-Resistant Bacteria Dissemination in the Wildlife, Livestock, and Water of Maiella National Park, Italy

**DOI:** 10.3390/ani13030432

**Published:** 2023-01-27

**Authors:** Camilla Smoglica, Alberto Vergara, Simone Angelucci, Anna Rita Festino, Antonio Antonucci, Fulvio Marsilio, Cristina Esmeralda Di Francesco

**Affiliations:** 1Department of Veterinary Medicine, University of Teramo, Loc. Piano D’Accio, 64100 Teramo, Italy; 2Wildlife Research Center, Maiella National Park, Viale del Vivaio, 65023 Caramanico Terme, Italy

**Keywords:** drug resistance, livestock, wild animals, water, *Escherichia coli*, *Enterococcus*, *Streptococcus*, *Aeromonas*, one health

## Abstract

**Simple Summary:**

Antibiotic resistance is a global problem that involves humans, animals and the environment. To counter this threat, a multidisciplinary strategy that addresses human, animal, and environmental health is needed. This approach, called One Health, was applied in this study involving wild and domestic species living in the Maiella national park (Italy). Feces and water samples from the same areas were collected to identify gram-positive and gram-negative bacteria and to define their susceptibility to several antibiotics. The results of this study showed several bacterial species with interesting resistance profiles, and in some cases, resistance against antibiotics that are critically important for human medicine. The sharing of environments between wild and domestic animals was confirmed as a key factor in antibiotic resistance dissemination at the human–animal–environment interface.

**Abstract:**

Antimicrobial resistance (AMR) is a global health concern that has been linked to humans, animals, and the environment. The One Health approach highlights the connection between humans, animals, and the environment and suggests that a multidisciplinary approached be used in studies investigating AMR. The present study was carried out to identify and characterize the antimicrobial resistance profiles of bacteria isolated from wildlife and livestock feces as well as from surface water samples in Maiella National Park, Italy. Ecological and georeferenced data were used to select two sampling locations, one where wildlife was caught within livestock grazing areas (sympatric group) and one where wildlife was caught outside of livestock grazing areas (non-sympatric group). Ninety-nine bacterial isolates from 132 feces samples and seven isolates from five water samples were collected between October and December 2019. The specimens were examined for species identification, antibiotic susceptibility and molecular detection of antibiotic resistance. Forty isolates were identified as *Escherichia coli,* forty-eight as *Enterococcus* spp., eight as *Streptococcus* spp. and ten as other gram-negative bacteria. Phenotypic antibiotic resistance to at least one antimicrobial agent, including some antibiotics that play a critical role in human medicine, was detected in 36/106 (33.9%, 95% CI: 25–43) isolates and multidrug resistance was detected in 9/106 isolates (8.49%, 95% CI: 3.9–15.5). In addition, genes associated with antibiotic resistance were identified in 61/106 (57.55%, 95% CI: 47.5–67) isolates. The samples from sympatric areas were 2.11 (95% CI: 1.2–3.5) times more likely to contain resistant bacterial isolates than the samples from non-sympatric areas. These data suggest that drug resistant bacteria may be transmitted in areas where wildlife and livestock cohabitate. This emphasizes the need for further investigations focusing on the interactions between humans, wildlife, and the environment, the results of which can aid in the early detection of emerging AMR profiles and possible transmission routes.

## 1. Introduction

Antibiotic therapy is widely recognized as one of the most successful therapeutic solutions in the history of medicine [1] and has played an essential role in the development of further medical discoveries such as organ transplants and chemotherapy [2]. The golden age of antibiotics started in the 1940s and developed over four decades, with more than 40 antibiotics being discovered for clinical use [1]. During this period, a cycle of antibiotic discovery, use/overuse, and appearance of resistance caught on [3]. The effects of this cycle became apparent beginning in the 1990s, during which a decreased number of novel antibiotics were reported. This phase was called the “dry pipeline” in the field of antibiotic research since the antibiotics being introduced for therapy applications were modified or combined versions of previously used molecules [1]. Indeed, the WHO’s last analysis of the clinical antibacterial pipeline highlights that only 7/43 antibiotics meet one of their innovation criteria: absence of known cross-resistance, new target, new mode of action, or new class. In addition, only two of the seven molecules are active against critical, multidrug-resistant gram-negative bacterial pathogens [4]. Bacteria with acquired non-susceptibility to at least one agent in three or more antibiotic classes are defined as being multidrug resistant (MDR) and consequently are associated with few treatment options [4]. Losing the effectiveness of antibiotic therapy in health care means that we are quickly approaching the “post antibiotic era” [1]. Different factors contribute to the development of antimicrobial resistance (AMR), including the prolonged, cumulative, brief exposure or overuse of antibiotics in human and animal health as well as the demographic changes associated with urbanization and the discharge of antibiotic residues into the environment [5,6,7]. In this view, AMR is considered to be a complex global problem that impacts the health of humans, animals, and the environment [8,9].

Wildlife living in contact with domestic animals, such as livestock, or in anthropized environments in rural or urban areas, has recently been recognized as a potential indicator of AMR dissemination. In addition, wild animals have been described as potential reservoirs of resistant bacteria or resistance genes [10,11]. However, the available studies only focus on a few bacterial species (mainly *Escherichia coli* and *Salmonella* spp.) and wild animals (mainly wild birds and small mammals) [11,12]. In addition, no data that includes endangered species and protected areas are available in European countries. Therefore, studying wild species that are characterized by specific habitats and restricted ranges may provide additional information about the transmission dynamics of AMR.

Despite the large and growing literature on AMR in medical and veterinary settings, recent critical reviews highlight that most studies are mainly focused on gram-negative bacteria and that a One Health approach is necessary to address AMR at the human–animal–environment interface [12,13].

These topics highlight the need for multidisciplinary studies to provide additional information on the recognized framework of One Health in AMR. Therefore, this study is an attempt to employ ecological data and microbiological investigations to describe and characterize the antimicrobial resistance profiles of gram-positive and gram-negative isolates obtained from natural water sources and fecal samples of wild and domestic ruminants collected in different areas of the Maiella National Park (Italy).

## 2. Materials and Methods

### 2.1. Study Area

The study area is the Maiella National Park (MNP), which is a protected area (about 740 km^2^) in the Central Apennine Mountains of Italy. The park territory is classified into homogeneous territorial units that are characterized by different degrees of protection based on their environmental features and the activities that are permitted there. In addition, the MNP is home to several species of mammals that are relevant at both the national and international level and listed in the Habitats Directive (92/43/EEC) and that coexist with local livestock facilities.

### 2.2. Sampling Design

The Apennine chamois (*Rupicapra pyrenaica ornate*), red deer (*Cervus elaphus*), sheep, goats, and cattle were selected as targeted species for sampling activities because they were either the most representative or the most peculiar species of ruminants in the study area.

The distribution areas of the wild and domestic animals were determined using georeferenced data and monitoring activities carried out by the technical staff of MNP. In detail, the size and distribution of the Apennine chamois population were defined using a census block technique over the last twenty years and the demographic structure and territory occupancy were described using data from GPS-GSM collars, ear tags, and direct visual observations that were collected during the last ten years [12].

The red deer population was characterized by estimating the minimum number of reproductive males during the rutting season and using data on population demographic structure that were established from recurrent visual observations during all the seasons [14]. Finally, domestic animal farms that required a specific authorization to use the grazing lands received an area that was defined by GPS coordinates and specific to each livestock group [14].

Based on these data, a first sampling area including sympatric populations (100 Apennine chamois coexisting with a farm of 120 goats, and 50 red deer coexisting with a farm of 300 sheep) and a second sampling area involving non-sympatric populations (70 cattle, 210 goats, and 100 Apennine chamois) were identified in the different territories of the MNP (Figure 1).

The specific sample collection sites were selected based on the GPS coordinates of the livestock grazing lands and the movements and distribution of wildlife populations. From October to November 2019, a total of 132 fecal samples (48 from sympatric areas and 84 from non-sympatric areas) from red deer, Apennine chamois, and extensive livestock were regularly collected in the morning. The collection of the samples was carried out as previously described, recovering at least 25 g for each specimen [14,15]. The samples were grouped into 33 fecal pools, each containing 4 fecal samples, and were stored at a temperature of +4 °C.

Additionally, 5 water samples (3 from sympatric areas and 2 from non-sympatric areas) were obtained by collecting water from the natural sources available in the sampling areas using 1 L sterile water bottles (Figure 1). The laboratory investigations were carried out within 24 h after the collection of the samples.

### 2.3. Bacteria Isolation and Antibiotic Susceptibility Test

Twenty-five grams of each sample was homogenized in 250 mL of buffered peptone water (BPW) using Stomacher^®®^ for 2 min.

In the case of water samples, bacterial cells were collected by 100 mL centrifugation (3000× *g* for 10 min), and the pellet was re-suspended with 5 mL of BPW.

Isolates were obtained by a preliminary non-selective enrichment of fecal and water samples in BPW (24 h at 37 °C), followed by subculture using the streak plating technique on MacConkey agar (Liofilchem, Italy) at 37 °C for 18–24 h and Slanetz–Bartley agar (Liofilchem, Italy) at 37 °C for 48 h. From each plate, 1 or 2 representative morphological colonies that were adequately isolated from other microorganisms were selected to obtain pure sub-colonies [14]. The species identification of colonies and the antimicrobial susceptibility test were performed using a Vitek 2 system (Biomerieux, France) and MIC Test strip (Liofilchem, Italy). When possible, the European Committee on Antimicrobial Susceptibility Testing (EUCAST) breakpoints were applied [16]. In cases where the EUCAST breakpoints were not available for the antibiotics and bacteria investigated, CLSI breakpoints were used instead [17].

The antimicrobial susceptibility patterns of gram-negative isolates were determined for 14 antimicrobials: ampicillin, piperacillin/tazobactam, cefotaxime, ceftazidime, ertapenem, meropenem, amikacin, gentamicin, ciprofloxacin, tigecycline, tetracyclines, nitrofurantoin, colistin, and trimethoprim/sulfamethoxazole. Susceptibility testing of *Enterococcus* spp. was performed for vancomycin, teicoplanin, linezolid, quinupristin/dalfopristin, gentamicin, kanamycin, streptomycin, ciprofloxacin, levofloxacin, daptomycin, tetracyclines, tigecycline, and nitrofurantoin. Finally, the susceptibility of *Streptococcus* spp. to ampicillin, benzylpenicillin, cefotaxime, ceftriaxone, chloramphenicol, clindamycin, erythromycin, gentamycin, levofloxacin, linezolid, moxifloxacin, rifamycin, teicoplanin, tetracycline, tigecycline, and trimethoprim/ sulfamethoxazole was evaluated. The antibiotics were selected by considering the main antimicrobial classes used in the livestock from the study areas and including some of the molecules that are critically important for human medicine.

### 2.4. Detection of Antibiotic Resistance Genes

In gram-negative isolates, genes related to resistance to beta lactams [*bla*TEM*, bla*SHV*, bla*CTX-M*, bla*CMY-1*, bla*CMY-2], carbapenems [*IMP, OXA-48 like, NDM, KPC*], colistin [*mcr-1, mcr-2, mcr-3, mcr-4, mcr-5*], tetracyclines [*tet*A*, tet*B*, tet*C*, tet*L*, tet*M*, tet*K], sulfonamides [*sul*1*, sul*2*, sul*3], and aminoglycosides [*aac*C1, *aac*3, *aac*A4, *aph*A6, *arm*A, *rmt*B, *rmt*C, *rmt*F] were screened by PCR, as previously described (Appendix A).

Concerning gram-positive isolates, PCR protocols were carried out as previously reported by other authors in order to investigate the presence of genes encoding resistance to quinupristin/dalfopristin [*vgA, msr*C*, Vat*D*, vgb*B*, vgb*A*, erm*B*, vat*E], vancomycin [*van*A*, van*B*, van*C1*, van*C2*, van*D*, van*G*, van*M*, van*N], linezolid [*cfr, cfr*B*, cfr*D*, optr*A*, poxt*A], nitrofurantoin [*nfs*A*, nfs*B], tetracyclines [*tet*A*, tet*B*, tet*C*, tet*L*, tet*K*, tet*M], macrolides [*erm*A, *erm*B, *erm*C, *erm*TR, *mef*A/E], quinolones [*gyr*A], and beta lactams [PBP1a, PBP2x, PBP2b] (Appendix A).

The details of the primer sequences and fragment sizes are specified in Appendix A.

### 2.5. Statistical Analysis

Descriptive analysis using a 95% confidence interval was performed using the standard statistical software packages provided by Stata [18]. The calculations for the relative risk (RR) of resistant bacteria detected in the different investigated areas/populations were performed in EpiSheet [19].

## 3. Results

A total of 106 isolates were selected from feces (*n* = 99) and water (*n* = 7). Table 1 summarizes all the recovered bacterial strains and their distribution in areas of sympatric or non-sympatric animals.

In detail, thirty-four isolates were collected from areas with sympatric animals and 72 isolates from areas with non-sympatric animals. The most frequent isolates were *Enterococcus* spp. (*n* = 48) and *E. coli* (*n* = 40).

Overall, thirty-six isolates (36/106, 33.9%, 95% CI: 25–43.8) showed phenotypic resistance to at least one antibiotic, and eleven isolates (11/106, 10.3%, 95% CI: 5.2–17.8) were classified as MDR bacteria showing resistance to at least three different classes of antibiotics. In addition, twenty-five isolates (25/106, 23.5%, 95% CI: 15.8–32.8) were found to be resistant to critically important antibiotics (CIAs) for human medicine. In detail, these isolates showed resistance to colistin (10/106, 9.4%, 95% CI: 4.6–16.6), linezolid (10/106, 9.4%, 95% CI: 4.6–16.6), vancomycin (3/106, 2.8%, 95% CI: 0.5–8), meropenem (3/106, 2.8%, 95% CI: 0.5–8), and ertapenem (3/106, 2.8%, 95% CI: 0.5–8). Four isolates (4/106, 3.7%, 95% CI: 1–9), including *E. coli, E. faecium, Ent. cloacae complex*, and *Str. mutans*, were identified as being resistant to two CIAs.

The antimicrobial susceptibility test of enterococci showed resistance to quinupristin/dalfopristin (8/48, 16.67%, 95% CI: 7.4–30.22), linezolid (8/48, 16.67%, 95% CI: 7.4–30.22), tetracycline (4/48, 8.33%, 95% CI: 2.31–19.98), vancomycin (1/48, 2.08%, 95% CI: 2.31–19.98), and teicoplanin (1/48, 2.08%, 95% CI: 2.31–19.98) as reported in Appendix A. Multidrug resistance to three or more different families of antibiotics was observed in one *Enterococcus faecium* and three *Enterococcus gallinarum*, as reported in Table 2. The resistance genes that were detected are described in Table 3. In detail, the quinupristin/dalfopristin resistance gene *msr*C was detected in 20 isolates, including six phenotypically resistant and 13 susceptible enterococci as well as one *E. faecalis* that was intrinsically resistant to this antibiotic. All the strains that were resistant to linezolid harbored the *cfr*D gene. Regarding *E. gallinarum*, a multiple positivity for *tet*M + *tetB* and *tet*M *+ tet*B *+ tet*L genes was found in one strain that was tetracycline susceptible and two that were tetracycline resistant, while two other tetracycline-resistant *E. gallinarum* isolates harbored only the *tet*M gene. None of the vancomycin resistance genes were amplified except for *van*C1and *van*C2, which were observed in 10 intrinsically resistant enterococci (*E. gallinarum* and *E. casseliflavus*) and in one susceptible isolate of *E. faecium*.

Analysis of the antimicrobial susceptibility of *E. coli* showed resistance to tetracycline (6/40, 15%, 95% CI: 5.71–29.83), ampicillin (3/40, 7.5%, 95% CI: 1.57–20.38), trimethoprim/sulfamethoxazole (2/40, 5%, 95% CI: 0.61–16.91), ceftazidime (1/40, 2.5%, 95% CI: 0.06–13.15), and meropenem (1/40, 2.5%, 95% CI: 0.06–13.15). One isolate was found to be MDR, as described in Table 2. The resistance genes that were detected in phenotypically resistant isolates are reported in Table 3.

Four resistant isolates of *Streptococcus* spp. (4/8, 50%, 95% CI:15.7–84.2) were identified. In detail, the strains were found to be resistant to ampicillin (2/8, 25%, 95% CI: 3.1–65), benzylpenicillin (2/8, 25%, 95% CI: 3.1–65), cefotaxime (4/8, 50%, 95% CI: 15.7–84.2), ceftriaxone (4/8, 50%, 95% CI: 15.7–84.2), clindamycin (4/8, 50%, 95% CI: 15.7–84.2), erythromycin (2/8, 25%, 95% CI: 3.1−65), linezolid (2/8, 25%, 95% CI: 3.1–65), vancomycin (2/8, 25%, 95% CI: 3.1–65), and tigecycline (2/8, 25%, 95% CI:3.1–65). Three MDR isolates were identified, and the profiles of resistance are reported in Table 2.

The remaining gram-negative bacterial species were found to be resistant to piperacillin/tazobactam (2/10, 20%, 95% CI: 2.5–55.6), cefotaxime (2/10, 20%, 95% CI: 2.5–55.6), ceftazidime (2/10, 20%, 95% CI: 2.5–55.6), amikacin (3/10, 30%, 95% CI: 6.6–65), meropenem (3/10, 30%, 95% CI: 6.6–65), ertapenem (3/10, 30%, 95% CI: 6.6–65), tigecycline (1/10, 10%, 95% CI: 0.2–44.5), and nitrofurantoin (1/10, 10%, 95% CI: 0.2–44.5).

Based on the PCR screening, sixty-eight isolates (68/106, 64.1%, 95% CI: 54.2–73.2) harbored at least one resistance gene related to the antibiotics being investigated in the present study. Additionally, forty-three isolates (43/106, 40.5%, 95% CI: 31–50.5) were found to carry at least one resistance gene related to CIAs (Table 3).

Finally, the RR analysis showed that bacteria isolated in areas with sympatric animals were 2.11 (95% CI:1.2–3.5, *p* = 0.0048) times more likely to be phenotypically resistant than bacteria isolated from non-sympatric areas.

## 4. Discussion

The present study investigated antibiotic resistance profiles in gram-positive and gram-negative bacteria from fecal samples of wild and domestic animals with different ecological patterns and from natural water samples related to the areas of study. The materials and methods were designed so that a One Health approach could be applied, as previously suggested in critical reviews on AMR [13,20]. Indeed, a multidisciplinary, multisector approach was strongly recommended to address health threats at the human–animal–environment interface [13,21]. In this view, rather than conducting a retrospective analysis of bacteria obtained with opportunistic sampling as has been done in other studies [22,23,24,25,26], the sampling procedures in this study were defined by considering the ecological features of the investigated animals in order to link ecology and antibiotic resistance.

The results allowed us to describe the AMR profiles in fecal and water samples collected from areas with different land use and highlight a 2.11 (95% CI: 1.2–3.5) times greater relative risk of resistant bacteria in areas with sympatric animals in comparison to areas of non-sympatric animals. Indeed, sharing an environments represents a key point of AMR spread between different ecological niches, as previously suggested in studies on baboons, buffalo and zebra that did not use georeferenced data [27].

Previous AMR studies using wildlife, livestock, or environmental samples have strongly focused on bacterial species relevant as human foodborne pathogens (i.e., *E. coli*, *Salmonella* spp, *Campylobacter* spp.) [12,13,28,29] or bacteria selected as valuable indicators of environmental contamination based on their adaptability and genome plasticity [30] such as *Enterococcus* spp.

These bacteria have been extensively investigated in previous studies [13,14,15,23,31], although combined investigations of multiple bacterial species at the wildlife–livestock interface are not widespread. *E. coli* and enterococci have been analyzed at the wildlife–livestock interface in buffalo, impala, wildebeest, zebra, and domestic cattle in Tanzania and Zambia [27,30,31]. In addition, similar studies on *Campylobacter* and *Salmonella* spp. have been conducted in sympatric livestock and wild species in Spain and California [32,33].

Although most previous studies focused on a single targeted bacterial species [13,14,15,23,31], recent critical reviews suggest that a wide range of bacteria should be investigated because emerging and interesting resistance profiles may be related to species of bacteria belonging to different settings [10,13]. In this view, the present study was carried out to obtain data on both gram-positive and gram-negative bacteria.

Enterococci and *E. coli* were the most identified bacterial species in the present study. In detail, resistant isolates of these bacterial species were obtained from fecal samples of non-sympatric domestic animals and sympatric wild and domestic animals.

In addition, one resistant isolate of *Morganella morganii* was identified in fecal samples of non-sympatric domestic animals, while one resistant isolate of *Aeromonas sobria* and two resistant isolates of *Pseudomonas mendocina* and *Enterobacter cloacae complex* were detected from water samples of both investigated areas.

Resistant isolates of *Morganella morganii* were previously reported in swine, poultry, cattle, dolphins, sea turtles, and animal products [34,35,36,37,38]. It is considered to be an opportunistic pathogen that may potentially cause fatal systemic infection, especially in nosocomial environments and in frail human patients such as young children or people with immune deficiency [39]. Considering the available data, resistant isolates of this bacterial species is recognized as a new clinical treatment challenge in human medicine [39].

*Aeromonas* spp., *Pseudomonas* spp., and *Enterobacter* spp. were previously isolated in veterinary settings and environmental sources and, similar to *Morganella morganii*, may have serious implications for human medicine [40,41,42,43].

In this study, resistant isolates of *Streptococcus* spp. were identified in fecal samples of sympatric wild and domestic animals and in non-sympatric domestic animals. In previous studies, *Streptococcus uberis* was found to be associated with clinical mastitis in dairy herds, *Streptococcus thoralthesis* and *Streptococcus alactolyticus* were described in mares, sows, rabbits, and in rare cases, humans [44,45,46,47,48,49,50], and *Streptococcus mutans* was isolated from dental caries or blood of human patients with infective endocarditis [51].

This study also detected isolates that were resistant to critically important antibiotics were in water and fecal samples from sympatric areas and in non-sympatric domestic animals. Critically important antibiotics include the more newly developed third generation cephalosporins, carbapenems, colistin, linezolid, and vancomycin drugs. These antibiotics are considered as last resources for the treatment of MDR infections in human medicine by the World Health Organization (WHO) [52]. Except for *E. coli* and enterococci [14,15,29], this study was the first to describe phenotypical and genetic resistance related to these antibiotics in gram-negative and gram-positive isolates detected at the wildlife–livestock interface. For example, although acquired linezolid resistance genes were previously reported in only two streptococcal species (*S. suis* and *S. gallolyticus*) [53], the *poxtA* gene was described in linezolid-resistant *S. uberis* obtained from sympatric chamois fecal samples in the present study. Similarly, although the NDM gene was previously only described in Europe in water sources distributed in the United Kingdom, Serbia, Switzerland, Ireland, Sweden, Spain, Belgium, and the Czech Republic [54], in this study, the carbapenems resistance gene NDM was detected in meropenem-resistant *Enterobacter cloacae complex.*

Concerning our other results, the resistances to tetracycline, ampicillin, and trimethoprim/sulfamethoxazole described in *E. coli* and *Enterococci* are in line with previous studies carried out in Zambia and Tanzania [27,30,31]. In addition, similar data were reported in studies focused on *E. coli* and enterococci from free-ranging wild mammals including red foxes and wild rabbits in Portugal and Norway [55,56,57] and lynxes and Iberian wolves in Spain [58]. Concerning other gram-negative bacteria detected in this study, similar phenotypic and genetic profiles were previously described in semi-aquatic wildlife in Spain and in white-tailed deer in the USA [59,60].

In contrast, our results regarding *Streptococcus* spp. represent a fraction of the limited data available on these bacteria in wildlife and at the wildlife–livestock interface. Indeed, the most recent studies in this area refer to the identification and characterization of isolates in chamois, without providing the AMR profiles [61].

The phenotypic and genetic analyses carried out in the present study represent a valid method for producing a snapshot of AMR in fecal and water samples, although the results could be improved by combining advanced techniques [13]. Two previous studies conducted in Italy and South Florida used metagenomic analysis to describe microbial communities and antibiotic resistance genes at the wildlife–livestock interface [62,63]. However, metagenomic analysis was still considered to offer low specificity with regards to detecting phenotypic traits [13,64].

Finally, the design of this study allowed us to identify bacteria that are resistant to critically important antibiotics included in the WHO’s list of global priority pathogens, such as vancomycin-resistant *Enterococcus faecium*, carbapenem-resistant *Pseudomonas* spp. and *Enterobacteriaceae*, and the third-generation cephalosporin-resistant bacteria *E. coli* and *Enterobacter* spp. [65]. Therefore, the data gathered from wildlife and environmental samples from areas with different levels of anthropic pressure could be included in future surveillance plans for monitoring antibiotic resistance. Such plans should not only consider antibiotics that are currently used in veterinary medicine, but also those that are clinically important drugs in human medicine.

## 5. Conclusions

This study reveals new insights about the occurrence of resistant bacteria in natural water sources as well as in rare species, such as the Apennine chamois, and other wild and domestic ruminants living in the distinctive ecological niche of Maiella National Park. The reported data suggest that the exposure of wildlife to antimicrobial resistance is largely determined by habitat use, particularly the level of interface with livestock and human activities. In addition, the identification of resistance profiles related to critically important antibiotics highlights the need to incorporate a One Health approach into global and local surveillance plans.

## Figures and Tables

**Figure 1 animals-13-00432-f001:**
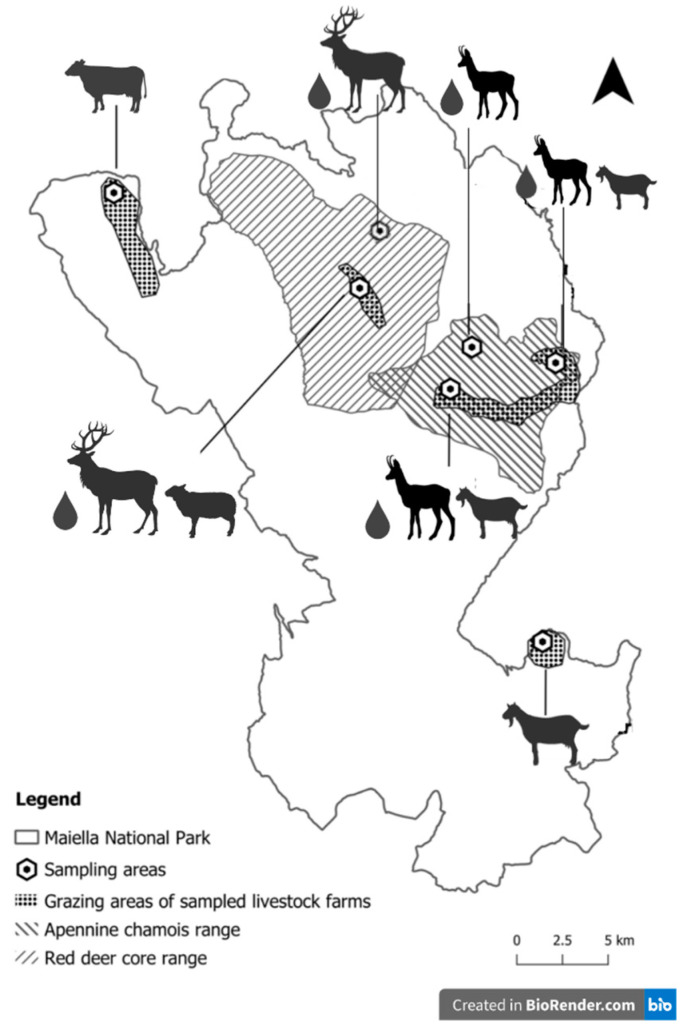
Distribution of the sampling areas in Maiella National Park.

**Table 1 animals-13-00432-t001:** Distribution of bacteria isolated from fecal and water samples collected in investigated areas of Maiella National Park where sympatric and non-sympatric animals were present.

Bacterial Species	Sample	Sympatric Animals Area	Non-Sympatric Animals Area	Total
*Acinetobacter baumannii*	W	0	1	1
*Aeromonas sobria*	W	1	0	1
*Citrobacter freundii*	D	0	1	1
*Enterobacter cloacae complex*	W	0	2	2
*Enterococcus casseliflavus*	AC, C, S	2	2	4
*Enterococcus faecalis*	AC, D, S	5	6	11
*Enterococcus faecium*	AC, C, D, G, S	4	8	12
*Enterococcus gallinarum*	C, D, G, S	3	12	15
*Enterococcus hirae*	AC, G, S	3	3	6
*Escherichia coli*	AC, C, D, G, S	14	26	40
*Klebsiella oxytoca*	W	1	0	1
*Klebsiella pneumoniae*	D	0	1	1
*Morganella morganii*	C	0	1	1
*Pseudomonas mendocina*	W	0	2	2
*Streptococcus alactolyticus*	S	1	0	1
*Streptococcus gallolyticus*	D	0	2	2
*Streprococcus mutans*	D	0	1	1
*Streptococcus thoraltensis*	C	0	1	1
*Streptococcus sanguinis*	C	0	2	2
*Streptococcus uberis*	AC	0	1	1
**Total**		**34**	**72**	**106**

AC: Apennine chamois; D: deer; C: cattle; G: goat; S: sheep; W: water.

**Table 2 animals-13-00432-t002:** Distribution of MDR bacteria isolated from fecal and water samples collected in investigated areas of Maiella National Park.

Strain	Source	Area	Multidrug Resistant Phenotypes	Number of Isolates
*Enterococcus gallinarum*	D, G	sympatric	LNZ QD TET	3
*Enterococcus faecium*	AC	sympatric	LNZ QD TEIC TET VAN	1
*Enterobacter cloacae complex*	W	non-sympatric	AK CTZ MER	1
*Escherichia coli*	G	non-sympatric	AMP SXT TET	1
*Pseudomonas mendocina*	W	non-sympatric	AK CTX ETP	1
*Pseudomonas mendocina*	W	non-sympatric	AK CTZ ETP	1
*Streptococcus mutans*	D	non-sympatric	AMP BEN CLIN CRO CTX LIN VAN TGC	1
*Streptococcus uberis*	AC	non-sympatric	AMP BEN CLIN CRO CTX ERY LIN TGC	1
*Streptococcus thoraltensis*	C	non-sympatric	CLIN CTX CRO VAN	1

AC: Apennine chamois; D: deer; C: cattle; G: goat; S: sheep; W: water; AMP: ampicillin; BEN: benzypenicillin; CLIN: clindamycin; CTX: cefotaxime; CRO: ceftriaxone; ERY: erythromycin; ETP: ertapenem; MER: meropenem; QD: quinupristin/dalfopristin; SXT: trimethoprim/sulfamethoxazole; TEIC: teicoplanin; TET: tetracycline; TGC: tigecycline; VAN: vancomycin.

**Table 3 animals-13-00432-t003:** Phenotypical and genotypical resistance profiles of bacteria isolated from fecal and water samples collected in investigated areas of Maiella National Park.

Bacteria	Sample	Antibiotics	Resistant Isolates	Gene Detected by PCR	
				Resistance Genes	Isolates
*Aeromon sobria*	W	MER, TZP/TAZ	1	*arm*A, *rmt*F	-
*Escherichia coli*	AC, S	AMP	3	*bla*CMY-2	1
				*tet*B	1
				-	1
	AC, S, D, C	CS	7	mcr-4, *bla*CMY-2	1
				mcr-4, *bla*TEM, *bla*CMY2	1
				mcr-4, t*et*B	1
				mcr-4, *bla*TEM	1
				mcr-4, *bla*CMY1, *bla*CMY2	3
	AC	TET	2	*tet*B, *bla*TEM, *bla*CMY2	1
				*tet*B	1
	AC, D	CS, TET	2	*tet*B, *bla*CMY2, mcr-4	1
				*tet*B, mcr-4	1
	AC	CS, CAZ, MRP	1	mcr-4, *bla*OXA-48, *bla*TEM, *bla*CMY1	1
	D	AMP, TET, SXT	1	-	-
	G	TET, SXT	1	-	-
*Enterococcus faecium*	D	QD	1	-	-
	AC	QD, LNZ, TEIC, VAN, TET	1	*msr*C, *Tet*B, *cfr*D	1
*Enterococcus faecalis*	S	LNZ	1	*cfr*D	1
*Enterococcus gallinarum*	D, S	QD, LNZ, TET	3	*cfr*D, *Tet*M	1
				*tet*B, *tet*M, *msr*C, *cfr*D	1
				*tet*B, *tet*M, *tet*L,*msr*C,*cfr*D	1
	G	QD, LNZ	3	*tet*B, *tet*M, *msr*C, *cfr*D	1
				*msr*C, *cfr*D	1
				*cfr*D	1
	D	QD, TET	1	*tet*M, *msr*C	1
*Streptococcus alactolyticus*	S	CTX, CRO, CLIN	1	*gyr*A	1
*Streptococcus mutans*	D	AMP, BEN, CTX, CRO, CLIN, LIN, TGC, VAN	1	*erm*A, *gyr*A	-
*Streptococcus thoraltensis*	C	CTX, CRO, CLIN, ERY, VAN	1	*van*C2, *van*G	1
*Streptococcus uberis*	AC	CTX, CRO, CLIN, LIN, AMP, BEN, ERY, TGC	1	*erm*B, *gyr*A, *par*C, *poxt*A, PBP2b	
*Enterobacter cloacae complex*	W	CTX, ETP, MER, TZP/TAZ	1	*bla*CTX-M, *bla*TEM, *bla*NDM, aaC1, *arm*A, aphA6, *rmt*F	1
	W	AK, CTZ, MER	1	*bla*CTX-M, *bla*TEM, *bla*NDM, *rmt*B, *arm*A, *aph*A6	1
*Pseudomonas mendocina*	W	AK, CTX, ETP	1	*bla*CTX-M	1
	W	AK, CTZ, ETP	1	*bla*CTX-M, *bla*CMY-1	1
*Morganella morganii*	C	NIT TGC	1	aaCA4, *nfs*A, *rmt*F, *rmt*B	1

AC: Apennine chamois; D: deer; C: cattle; G: goat; S: sheep; W: water; AMP: ampicillin; AK: mikacin; BEN: benzypenicillin; CS: colistin; CLIN: clindamycin; CTX: cefotaxime; CRO: ceftriaxone; ERY: erythromycin; ETP: ertapenem; MER: meropenem; NIT: nitrofurantoin; QD: quinupristin/dalfopristin; SXT: trimethoprim/sulfamethoxazole; TEIC: teicoplanin; TET: tetracycline; TGC: tigecycline; TZP/TAZ: piperacillin/tazobactam; VAN: vancomycin.

## Data Availability

The data presented in this study are contained within the article and further information will be made available upon reasonable request to the corresponding author.

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
