# Peer review of "Antibiotic-Resistant Bacteria Dissemination in the Wildlife, Livestock, and Water of Maiella National Park, Italy"

_animals, 2023, doi:10.3390/ani13030432_

Round 1

Reviewer 1 Report

Review Report of the manuscript ID: animals-2160645

Title: Antibiotic resistant bacteria dissemination in wildlife, livestock and water of Maiella National Park, Italy

Summary

This manuscript characterizes antimicrobial resistance profiles of bacterial isolates in faecal samples of wild and domestic animals, and in water samples collected in a protected region located at the central Apennine Mountains in Italy. In this study, bacteria were isolated and identified. Next, antibiotic susceptibility test was performed, and a molecular analysis was conducted to detect antibiotic resistance genes. The study was conducted in two different areas allowing the comparison of the results between populations of the two areas.

General concepts

Globally, this manuscript is clearly written. However, some issues need a more detailed clarification. Furthermore, check spelling is needed throughout the manuscript (e.g. line 271, line 292) and the manuscript must be revised by a native English speaker.

Concerning references and citations, all the manuscript contains 65 references and 61,5% were published recently (from 2018). But citations must be checked through the manuscript (e.g. line 62; line 319).

Specific comments

Summary:

Is a little confusing. It should be improved and organized objectively.

Abstract :

Line 29-30: how many samples were analysed? (This issue must be also clarified in M&M)

Line 32: Is it necessary to refer statistical software in the abstract?

Line 38-39: sympatric and non-sympatric should be mentioned before to contextualize the terminology

Keywords:

Acinetobacter is included but it is mentioned only once (table 1) in all the manuscript. Is it important? Why is it not mentioned in discussion? Acinetobacter baumanii was isolated from a water sample. Does it have impact on animal or in human health?

Spelling – Streptococcus instead of Streprococcus

Introduction

The concept of multidrug resistance (MDR) should be explained in the introduction.

Uniformise Gram negative / Gram-negative and Gram positive / Gram-positive throughout the manuscript.

M&M

Line 89 - How / why were these species selected for the study?

Line 96, 99, 102 – I don’t understand why the references [12,13] are cited here.

Faecal samples were collected from soil. However, authors refer that only freshly faeces were collected. How many faecal samples were collected in each sampling area? How were the collection sites defined in each area? And were the samples collected in a specific period of the day (e.g. morning). It should be clarified, as well as the amount of faeces collected.

Furthermore, it is mentioned that sampling occurred during October and November. Can the season influence the results?

Line 112 – Why were samples pooled? Was it possible to determine the origin of each sample (species)?

Line 114 – “refrigerator temperature” please specify in an objective way.

Line 115 – Please specify the method of collection, volume and characteristics of the container used.

Soil samples were not collected. Why? Wouldn’t it be an added value in this study?

Line 120 – Suggestion “Bacteria isolation and antibiotic susceptibility test”

Line 123 – which volume of water was used for centrifugation?

Line 126 – which method was used for inoculation of culture media.

The aim of the study was to characterize the antimicrobial resistance profiles of Gram positive and Gram negative bacteria. However, culture was performed on MacConkey Agar (selective for Gram negative) and on Slanetz-Bartley agar (selective medium for isolation of enterococci). So, the spectrum of isolated bacteria is not as wide as it seems.

Which criteria were used to select isolates?

Line 135-144 – which criteria were used to select antibiotics? Suggestion: antibiotics were selected according to ….

Line 153 – Please specify the reference “by other authors”

Results

Line 166 – “106 isolates were selected”

Line 173 – “The most frequent isolates…

Lines 251-253 – Is it statistically significant?

Table 3 – Please clarify “N of R isolates” and “N of isolates” in the head of the table

It was found that 7 isolates were resistant to colistin. Is it an important finding?

Some isolates showed a resistant profile to different antibiotics, but molecular analysis did not evidence the presence of genes of resistance. Why did it happen?

Discussion

Line 279 – What do you mean by RR? It is not clearly mentioned in the results.

Needs an improvement.

Is the number of tested samples and the number of isolates studied representative?

Which are the strengths of your study?

Did you find differences in resistance profiles between sympatric and non-sympatric populations?

Lines 367-370 – This sentence should be clarified in an objective way.

Conclusions

An improvement is needed to highlight objectively the main findings of this study.

Author Response

Comments and Suggestions for Authors

Review Report of the manuscript ID: animals-2160645

Title: Antibiotic resistant bacteria dissemination in wildlife, livestock and water of Maiella National Park, Italy

Summary

This manuscript characterizes antimicrobial resistance profiles of bacterial isolates in faecal samples of wild and domestic animals, and in water samples collected in a protected region located at the central Apennine Mountains in Italy. In this study, bacteria were isolated and identified. Next, antibiotic susceptibility test was performed, and a molecular analysis was conducted to detect antibiotic resistance genes. The study was conducted in two different areas allowing the comparison of the results between populations of the two areas. 

General concepts

Globally, this manuscript is clearly written. However, some issues need a more detailed clarification. Furthermore, check spelling is needed throughout the manuscript (e.g. line 271, line 292) and the manuscript must be revised by a native English speaker.

 As suggested by Reviewer, the manuscript was revised and checked.

Concerning references and citations, all the manuscript contains 65 references and 61,5% were published recently (from 2018). But citations must be checked through the manuscript (e.g. line 62; line 319).

 Done

Specific comments

Summary:

Is a little confusing. It should be improved and organized objectively.

 As suggested by Reviewer, the Summary was improved.

Abstract :

Line 29-30: how many samples were analysed? (This issue must be also clarified in M&M)

Done in both Abstract and Materials and Methods sections.

Line 32: Is it necessary to refer statistical software in the abstract?

According to Reviewer’s suggestion, the details of Statistical software were removed from the abstract.

Line 38-39: sympatric and non-sympatric should be mentioned before to contextualize the terminology.

Done

Keywords:

Acinetobacter is included but it is mentioned only once (table 1) in all the manuscript. Is it important? Why is it not mentioned in discussion? Acinetobacter baumanii was isolated from a water sample. Does it have impact on animal or in human health?

Despite Acinetobacter baumanii can be considered an opportunistic pathogen for humans, the isolate obtained from our study showed susceptibility to all tested antibiotics. Precisely for this reason, it wasn’t’ in the discussion section and as suggested it was removed from keywords.

Spelling – Streptococcus instead of Streprococcus

Done. We apologize for the typo.

Introduction 

The concept of multidrug resistance (MDR) should be explained in the introduction.

Done (lines 70-73)

Uniformise Gram negative / Gram-negative and Gram positive / Gram-positive throughout the manuscript.

Done

M&M

Line 89 - How / why were these species selected for the study?

The most peculiar or representative ruminants species living in study areas were selected in order to obtain valuable data showing AMR diffusion in the territory. The text was consequently modified.

Line 96, 99, 102 – I don’t understand why the references [12,13] are cited here.

The reference 12-13 (now changed in 14-15) were cited because they described the procedure applied to obtain ecological and georeferenced data and fresh samples. In this view, the text concerning of sampling activities was removed.

Faecal samples were collected from soil. However, authors refer that only freshly faeces were collected.

How many faecal samples were collected in each sampling area? How were the collection sites defined in each area?

And were the samples collected in a specific period of the day (e.g. morning). It should be clarified, as well as the amount of faeces collected.

The details requested by Reviewer were added in Lines 130-144.

Furthermore, it is mentioned that sampling occurred during October and November. Can the season influence the results?

The choice of autumn season has allowed to collect and recover fresh samples improving the sensitivity of microbiological analysis, but the Authors have not additional available data regarding the influence of seasons.

Line 112 – Why were samples pooled? Was it possible to determine the origin of each sample (species)?

 The samples were pooled based on the species and the site of origin, in order to obtain a set of bacteria of each animal population.

Line 114 – “refrigerator temperature” please specify in an objective way.

Done

Line 115 – Please specify the method of collection, volume and characteristics of the container used.

Done

Soil samples were not collected. Why? Wouldn’t it be an added value in this study?

 We agree with Reviewer and we will consider the suggestion for future investigations.

Line 120 – Suggestion “Bacteria isolation and antibiotic susceptibility test”

Done

Line 123 – which volume of water was used for centrifugation?

Added.

Line 126 – which method was used for inoculation of culture media.

The plating technique was specified.

The aim of the study was to characterize the antimicrobial resistance profiles of Gram positive and Gram negative bacteria. However, culture was performed on MacConkey Agar (selective for Gram negative) and on Slanetz-Bartley agar (selective medium for isolation of enterococci). So, the spectrum of isolated bacteria is not as wide as it seems.

Considering that the analyzed samples are mainly feces the investigations were focused on the bacterial species (Enteobacteriaceae and enterococci) widely recognized such as indicators of AMR in this kind of specimen. However, it was possible to identify other Gram-positive bacteria using the selective medium for enterococci, as reported in Table 1.

Which criteria were used to select isolates?

 As reported in Line 156, the morphology of colonies was used to select the isolates.

Line 135-144 – which criteria were used to select antibiotics? Suggestion: antibiotics were selected according to ….

Done

Line 153 – Please specify the reference “by other authors”

The related references are specified in Supplementary Table 1.

Results

Line 166 – “106 isolates were selected”

 Done

Line 173 – “The most frequent isolates…

 Done

Lines 251-253 – Is it statistically significant?

 As suggested by reviewer, the statistical significance of RR value was highlighted.

Table 3 – Please clarify “N of R isolates” and “N of isolates” in the head of the table

 Done

It was found that 7 isolates were resistant to colistin. Is it an important finding?

The resistance to Critically Important Antibiotic such as colistin was remarked in discussion section (Lines 368-383).

 Some isolates showed a resistant profile to different antibiotics, but molecular analysis did not evidence the presence of genes of resistance. Why did it happen?

The absence of resistant genes may be re related to the detection limits of PCR protocols or other genetic markers not tested in our study may be involved.

Discussion 

Line 279 – What do you mean by RR? It is not clearly mentioned in the results.

Needs an improvement.

Done

Is the number of tested samples and the number of isolates studied representative?

We appreciated the valuable suggestion of the Reviewer; however, the aim of our study is to compare two different ecological conditions, with or without sympatry of animals, performing an observational study on the diffusion of the antimicrobial resistance and not a prevalence analysis. In this view the number of samples was calculated based on the size of populations under study and compatibly with the logistic challenges related to elusive species as Apennine chamois and red deer.

Which are the strengths of your study?

The novelty of our study was highlighted in Discussion and Conclusions sections improving the text based on the suggestions of Reviewers n. 1 and n.2.

Did you find differences in resistance profiles between sympatric and non-sympatric populations?

 The results didn’t show any relevant difference of the resistance profiles between sympatric and non-sympatric populations.

Lines 367-370 – This sentence should be clarified in an objective way.

 The Authors have modified the sentence in order to clarify the take home message.

Conclusions

An improvement is needed to highlight objectively the main findings of this study.

As suggested by Reviewer the conclusions were improved.

Reviewer 2 Report

This manuscript shows the results obtained from a study carried out in an Italian National Park on antimicrobial resistance in wildlife and livestock. Antimicrobial resistance is one of the biggest challenges facing modern medicine, and therefore research must be carried out from a One Health approach to obtain as much information as possible. In this context, the results of the present study have great value, however, it is necessary to improve the quality of the manuscript a little.

In my opinion, the introduction is really short. Although it is true that the problem of antimicrobial resistance and its importance at the global health level is well introduced, I missed reading about the importance of wildlife as carriers/spreaders of AMR in ecosystems and bioindicators of environmental contamination and the anthropization of ecosystems. I think the introduction should be expanded a bit, including one or two paragraphs on this topic, which justify the objectives of the research.

On the other hand, data is repeated in the results section. Normally, the data presented in tables are not included in the text, only a brief summary with the highlights. I think that the data in lines 195-202, 209-216, and 221-236 should be eliminated since they all appear in tables 2 and 3. To make the reading a little more pleasant, I recommend moving lines 237-246 to the beginning of the section on results and then moving on to the details.

On the other hand, I suggest eliminating from the discussion the information on the limitations of the study based on the WGS, NGS, and metagenomics. I think the authors are "throwing stones at their roof" and it overshadows all the work done and the results obtained.

For the conclusions, they are quite superfluous. I recommend the authors risk a little more with the conclusions and be more specific.

Finally, I have some small suggestions throughout the text:

L13. "Multidisciplinary strategies that address human, animal, and environmental health are needed to counter this threat" Rephrase this sentence.

L26. The abstract begins directly with the objective of the study. I recommend adding one or two introductory sentences.

L50. Please change pivotal to essential.

L71. Gram negative. Throughout the text, we read Gram negative and Gram-positive. Please decide which format you prefer and use the same format throughout the manuscript, hyphenated or unhyphenated.

L291. It looks like there is a double space after the point. The same happens on line 299.

L312. There is a parenthesis after reference 39.

L319. Please include the reference Martins et al, 2021 in the correct format.

L320. Please remove the italics from and.

L324. Please exchange specimens for samples.

L326. Please change includes to include.

L342. Zambia and Tanzania are countries. It is not necessary to specify it. Please delete the word countries.

L345-346. Other Enterobacteriaceae and non-Enterobacterioceae Gram-negative bacteria = Other Gram-negative bacteria. Please simplify the sentence. 

Author Response

Comments and Suggestions for Authors

This manuscript shows the results obtained from a study carried out in an Italian National Park on antimicrobial resistance in wildlife and livestock. Antimicrobial resistance is one of the biggest challenges facing modern medicine, and therefore research must be carried out from a One Health approach to obtain as much information as possible. In this context, the results of the present study have great value, however, it is necessary to improve the quality of the manuscript a little.

In my opinion, the introduction is really short. Although it is true that the problem of antimicrobial resistance and its importance at the global health level is well introduced, I missed reading about the importance of wildlife as carriers/spreaders of AMR in ecosystems and bioindicators of environmental contamination and the anthropization of ecosystems. I think the introduction should be expanded a bit, including one or two paragraphs on this topic, which justify the objectives of the research.

The introduction was improved following the suggestions of the Reviewer (lines 80-89)

On the other hand, data is repeated in the results section. Normally, the data presented in tables are not included in the text, only a brief summary with the highlights. I think that the data in lines 195-202, 209-216, and 221-236 should be eliminated since they all appear in tables 2 and 3. To make the reading a little more pleasant, I recommend moving lines 237-246 to the beginning of the section on results and then moving on to the details.

Done

On the other hand, I suggest eliminating from the discussion the information on the limitations of the study based on the WGS, NGS, and metagenomics. I think the authors are "throwing stones at their roof" and it overshadows all the work done and the results obtained.

In accordance with this suggestion we reorganized the paragraph removing the comment about the limitations of our study.

For the conclusions, they are quite superfluous. I recommend the authors risk a little more with the conclusions and be more specific.

The conclusions section was modified in accordance with both reviewers.

Finally, I have some small suggestions throughout the text:

L13. "Multidisciplinary strategies that address human, animal, and environmental health are needed to counter this threat" Rephrase this sentence.

Done

L26. The abstract begins directly with the objective of the study. I recommend adding one or two introductory sentences.

Done

L50. Please change pivotal to essential.

Done

L71. Gram negative. Throughout the text, we read Gram negative and Gram-positive. Please decide which format you prefer and use the same format throughout the manuscript, hyphenated or unhyphenated.

Done

L291. It looks like there is a double space after the point. The same happens on line 299.

Done

L312. There is a parenthesis after reference 39.

Done

L319. Please include the reference Martins et al, 2021 in the correct format.

Done

L320. Please remove the italics from and.

Done

L324. Please exchange specimens for samples.

Done

L326. Please change includes to include.

Done

L342. Zambia and Tanzania are countries. It is not necessary to specify it. Please delete the word countries.

Done

L345-346. Other Enterobacteriaceae and non-Enterobacterioceae Gram-negative bacteria = Other Gram-negative bacteria. Please simplify the sentence. 

Done

Round 2

Reviewer 1 Report

I have no more suggestions to present to the authors.